# TRPV1 Responses in the Cerebellum Lobules V, VIa and VII Using Electroacupuncture Treatment for Inflammatory Hyperalgesia in Murine Model

**DOI:** 10.3390/ijms21093312

**Published:** 2020-05-07

**Authors:** Chanya Inprasit, Yi-Wen Lin

**Affiliations:** 1College of Chinese Medicine, Graduate Institute of Acupuncture Science, China Medical University, Taichung 40402, Taiwan; chan-y-yaa@hotmail.com; 2Chinese Medicine Research Center, China Medical University, Taichung 40402, Taiwan

**Keywords:** inflammatory pain, TRPV1, electroacupuncture, ST36, cerebellum

## Abstract

Inflammatory pain sensation is an important symptom which protects the body against additional tissue damage and promotes healing. Discovering long-term and effective treatments for pain remains crucial in providing efficient healthcare. Electroacupuncture (EA) is a successful therapy used for pain relief. We aimed to investigate effects and mechanisms of Complete Freund’s Adjuvant (CFA)-inducing inflammatory pain in the cerebellum, and the inhibition of this inflammatory hyperalgesia using EA at Zusanli acupoint (ST36). The results display a significant increase in mechanical and thermal sensitivities in the CFA and CFA + SHAM groups, which was significantly reduced in the CFA+EA and CFA + KO groups. This evidence was substantiated in the protein levels observed using immunoblotting, and presented with significant escalations after CFA inducing inflammatory hyperalgesia in CFA and CFA + SHAM groups. Then, they were significantly attenuated by EA in the CFA + EA group. Furthermore, the CFA + transient receptor vanilloid member 1 (TRPV1)^−/−^ group indicated similar significant decreases of protein expression. Additionally, a concomitant overexpression in lobule VIa was also observed in immunofluorescence. These consequences suggest that CFA-induced inflammatory pain provokes modifications in cerebellum lobules V, VIa and VII, which can subsequently be regulated by EA treatment at the ST36 through its action on TRPV1 and related molecular pathways.

## 1. Introduction

Inflammatory pain is the outcome of pathological mechanisms, initiating from tissue injury [1]. The tissue damage releases endogenous biochemicals including the transient receptor vanilloid member 1 (TRPV1), which can activate nociceptors to propel the signals from peripheral afferent neurons to the spinal cord, dorsal root neurons and further to the brain [2]. TRPV1 is one of the six members in TRPV subfamily, being the first unit specified and the most commonly characterized with regard to research of nociceptive responses. TRPV1, a nonselective calcium-permeable cation channel, consents a neurotransmitter surge and further cascades the signal to the subsequent neurons or nucleus. TRPV1 is well-known for its inflammatory associations due to factors, such as its low pH (lower than 6.0) and high temperature (higher than 43 degrees Celsius) and its tendency to capably motivate the receptor by binding to a specific molecular site then increasing membrane depolarization [3,4]. These stimuli are an important part of the inflammatory mediators that are released and are responsible for the five typical signs of inflammation: pain, swelling, redness, increased heat, and loss of function [5]. Even though opioid drugs are optionally used for severe acute and cancer-related pain resultant from their rapid onset, prolonged effect, and excessive sedation, the recommendation of opioids as a first line treatment for pain was changed because of the large number of reported side effects and risks [6,7]. Compellingly, non-steroidal anti-inflammatory drugs (NSAIDs), such as aspirin, ibuprofen, and naproxen, demonstrate reduced risks compared to opioid drugs, although they also exhibit an array of negative side effects, including gastrointestinal bleeding (which is the most frequently reported), reduced renal blood flow, and deceased platelet function [8].

Traditional medicine worldwide conventionally utilizes and adapts natural methods to cure illnesses such as pain, which is the primary physiological response that warns the human body that something is amiss. In ancient times, Mesopotamia, Egypt, Greece, and Rome used opium for pain management, whereas in China, pain problems were treated through acupuncture as described in the Huangdi Neijing medical book, written over 3000 years ago [9]. Acupuncture has scientific evidence to prove the involvement of qi (energy) stimulation through meridians in accordance with Traditional Chinese Medicine (TCM) theory. Currently, this term of Qi can be classified as the nervous system in strictly scientific terms. It not only produces an analgesic sensation to local nociception as a trigger point treatment, but also in other parts of the body, which can be explained by the efficacy of acupuncture through the central nervous systems (CNS) [10]. Acupuncture can also be combined with various stimulating methods, such as lifting, thrusting, twisting, twirling, and other techniques. Today, the modern method of electroacupuncture (EA) is replacing traditional manual manipulations and can be controlled by manipulating the intensity and frequency of an electric current applied through a needle at the point of insertion. Different frequencies of EA stimulation activate different neurotransmitters. Low-frequency EA at 2 Hz to 4 Hz induces enkephalin release to activate the µ-receptor. By contrast, high-frequency EA (50 Hz to 100 Hz) stimulation releases dynorphins to activate κ-receptors [11]. However, one experiment demonstrated that EA at both 2 Hz and 100 Hz significantly relieved painful sensations, whereas no significant differences in terms of analgesic effects between these two EA frequencies were detected [12]. Moreover, different needle retention durations of acupuncture treatment demonstrated non-significant differences in terms of various signs and symptoms [13]. Zusanli acupoint (ST36), is positioned longitudinally three body inches below the knee joint, intersecting with the middle of the tibialis anterior muscle, and is most commonly used for the treatment of pain, lower limb issues, and gastrointestinal disorders. It can also be used to treat diseases that manifest along the stomach meridian [14].

The cerebellum, a part of the CNS, is located posterior to the brainstem and inferior to the cerebrum. Characteristically, the cerebellum has the function of coordinating movement and learning based upon its specific areas [15]. The cerebellum communicates to different parts of the body, including both sensory and motor pathways, through the role of Purkinje cells. The Purkinje cells and dendrites are the largest neurons in the cerebellum which transmit signals between the cerebellar nuclei and cerebellar cortex via ion channels, including Na^+^ and K^+^. They also contain voltage-gated Ca^2+^ channels that can be triggered by depolarization from climbing or parallel fibers in the cerebellum, and intracellular Ca^2+^ store release [16]. However, the cerebellum has received little attention from researchers in the field of nociception, although it has been termed “little brain” and is responsible for a multitude of important physiological responses. Moreover, Purkinje cells in the cerebellum contain voltage-gated Ca^2+^ channels that share an ion preference type with TRPV1 nociceptors [17,18]. The effect of EA treatment on the pain mechanism in the cerebellar functions remain unclear. Furthermore, the prescription of pain relief should be meticulously considered by skilled physicians to provide maximum effectiveness with minimum side effects for a variety of pain conditions and pathways. This study aimed to clarify the efficacy of EA treatment and modification of molecular substrates in the cerebellum under inflammatory pain conditions. Complete Freund’s adjuvant (CFA), a combination of components of heat-killed Mycobacterium tuberculosis and non-metabolizable oils (paraffin oil and mannide monooleate), has been frequently used for inflammatory pain stimulation in animal model laboratory research [6]. Therefore, we hypothesized that CFA injection causes alterations in the cerebellum due to inflammatory pain which can be relieved using EA at ST36 acupoint. These effects and mechanisms of inflammatory pain were also hypothesized to involve the TRPV1 receptor and Mitogen-Activated Protein Kinase (MAPK) pathways in the cerebellum.

## 2. Results

### 2.1. The Effect of EA Treatment on Mechanical and Thermal Behavior

The standard criteria of inflammatory pain in murine models are mechanical and thermal sensitivity examinations. The von Frey test is a mechanical sensitivity method that is frequently performed in pain experiments, whilst the Hargreaves’ method is used for thermal sensitivity analysis. The data on Day 0, considering the baseline of the experiment in all subjects, without any interventions applied, indicated no statistical significance in both the von Frey and Hargreaves’ tests among the five groups; control, CFA, CFA + EA, CFA + SHAM, and CFA + KO. After the CFA inflammatory pain induction, mechanical sensitivity results on Day 1 displayed statistical decreases in subjects from the CFA, CFA + EA, CFA + SHAM, and CFA + KO groups when compared to the control group. However, the CFA+KO group displayed a propensity to a lesser state of inflammatory pain, as this group, whilst significantly different from the control group, was also significantly different from the other three groups, CFA, CFA + EA, CFA + SHAM. This parallel tendency was similarly observed in the thermal sensitivity results, in that the analyzed data from the CFA + KO group only indicated a significant difference when compared to the CFA, CFA + EA and CFA + SHAM groups. However, no significant difference was observed when comparing the CFA + KO group to the control group. This suggests that the thermal sensitivity of TRPV1 KO mice is expectantly reduced according to the role of TRPV1 in thermal hyperalgesia. On Day 2, after the first treatment session, the von Frey filament outcomes were significantly escalated in the CFA + EA group after receiving EA treatment at ST36 2 Hz/20 min when compared to the CFA and CFA + SHAM groups, whilst the difference between the CFA + EA and control group was also still statistically significant. The data of the CFA + KO group, which lacks the TRPV1 nociceptor, also reveal a distinguishable increase when compared to the baseline of the control group. Moreover, in the Hargreaves’ thermal sensitivity test, significant increases are shown in paw withdrawal latency in the CFA + EA and CFA + KO groups when compared with both no treatment and placebo treatment groups, but not when contrasted to the control group. Continuously, the reductions in both examinations were significantly attenuated in the CFA + EA and CFA + KO groups when contrasted to the CFA and CFA + SHAM groups. No significant variance is found among the control, EA treatment and TRPV1 gene deletion groups on Day 3 (Figure 1).

### 2.2. The Effect of EA Treatment on Nociceptor and Its Downstream in the Cerebellum Lobules V, VI and VII

After the inflammatory pain was confirmed according to the behavioral changes observed in the mechanical von Frey test and thermal Hargreaves’ test, the brain samples were collected to detect the protein modification with regards to the cerebellum lobules V, VI and VII. First, in the cerebellum lobule V (Figure 2) it was found that the pain receptors of TRPV1, acid sensing ion channel subtype 3 (ASIC3), Voltaged-gated Sodium (Nav)1.7, Nav1.8 and the receptor for advanced glycation end products (RAGE), displayed significant increases in the CFA and CFA + SHAM groups, which received CFA injection without any actual treatment in comparison to the standard of the control group. These increases were significantly diminished in the CFA+EA and CFA + KO groups which underwent EA treatment or had an absence of the TRPV1 receptor. The homologous tendencies were revealed in the density of pPI3K, pmTOR, pAkt and pERK, which are the principle pathways of MAPK. Moreover, the important molecules in inflammation, pPKAIIα and pPKCε, were also observed and presented significant decreases in the control, CFA + EA and CFA + KO groups when compared to the CFA and CFA + SHAM groups. The S100B protein was examined for its ability to act on the RAGE receptor stimulation. There are significant greater levels of S100B in the CFA and CFA + SHAM groups, however, the significantly minor levels were shown in the other three groups. Finally, the data of pNFκB and pCREB, which function inside the nucleus, indicate a similar trend as nociceptors in inflammatory-related molecule pathways.

Next, the overall cerebellum lobules VIa and VIb (Figure 3) were concurrently dissected due to the limitation of scope identification. Accordingly, the immunofluorescence technique was utilized in compensation for this weak point. The data of TRPV1 protein levels in the cerebellum total lobules VI displayed insignificant differences among the four wild-type groups, whilst a significant reduction was observed in the CFA + KO groups. There were no significant variances of Nav1.7, and Nav1.8 expression among the five groups of subjects, regardless of both wild-type and knockout mice. However, the ASIC3 and RAGE outcomes demonstrate slightly decreased protein density in the CFA + KO group in contrast to the other wild-type groups. The intensity of pPI3K, pmTOR, pAkt, pCREB, pPKAIIα and pPKCε also exhibit a non-statistical difference, as observed in the two Voltaged-gated Sodium (Nav) channel receptors. Nonetheless, there were compatible tendencies of pNFκB and S100B protein concentrations observed in the ASIC3 and RAGE graphs. Apart from other pain receptors and related molecules, the pERK results expose a divergent trend of statistical diminution in the CFA + KO group when compared with other groups.

In addition, we inspected the cerebellum lobules VII (Figure 4), investigating the aforementioned antibodies in lobules V and VI. The analogous tendencies of the intenseness of all 14 proteins between lobules V and VII were identified accordingly; CFA injection induced a significant elevation in the CFA and CFA + SHAM groups when compared to the painless control group. These significant decreases were observed in the inflammatory pain incitement CFA + EA group, which afterwards underwent an EA therapy, and also the CFA + KO group, which lacks the TRPV1 pain receptor. 

### 2.3. The Effect of EA Treatment on Protein Expression Alteration in the Cerebellum Lobules V, VIa, VIb and VII via Immunofluorescence Technique

Finally, the qualitative evidence in the cerebellum, with specificity to lobules V–VII, was qualified using the immunofluorescence technique, and is accordingly presented in Figure 5 and Figure 6. The images of the cerebellum lobule V displayed significant increases of TRPV1 and pNFkB protein staining in the CFA and CFA + SHAM groups, in contrast with the control, CFA + EA, and CFA + KO groups, which accordingly present a significant reduction in protein density. The similar tendencies of TRPV1 and pNFkB protein intensity were also found in the cerebellum lobules VIa and VII. Conversely, an insignificant variance of both TRPV1 and pNFkB protein levels among the five groups of subjects were discovered in the cerebellum lobule VIb area.

## 3. Discussion

Inflammatory pain, an unpredictable condition with challenging management, occurs by tissue destruction which stimulates the release of inflammatory mediators and further induces pain sensitivity through nociceptors such as TRPV1 [19,20]. TRPV1, a calcium ion channel, is distinguished for its involvement in inflammatory states due to its sensitivities to acidity and heat [21]. Inflammatory pain has been appealing in an epidemiological setting since the middle of the nineteenth century. To this end, a large variety of medications have been developed and used to reduce the incidence of pain. Currently, the research focus is aimed at increasing efficacy and decreasing the side effects of treatments such as opioid drugs and NSAIDs [22]. Shortly thereafter, the effectiveness of clinical acupuncture compared with western medicine treatment in inflammatory pain was studied and first published. The number of publications has been increasing both in clinical and animal research, alongside the development of modernized acupuncture techniques, since resultant effects display a beneficial impact in pain management with a markedly reduced number of side effects [23]. Both central and peripheral nervous systems are associated with inflammatory pain resulting in a plethora of developed treatment approached based on relative mechanistic action [24]. We induced CFA-induce inflammatory pain and then conducted the von Frey and Hargreaves’ examinations to firstly observe behavioral change in order to evaluate the success of inflammatory pain induction. After this, the cerebellum samples were obtained for immunoblotting and immunofluorescence analysis in order to determine modifications in the cerebellum among subject groups and to assess the role of the TRPV1 receptor in this condition and the respective efficacy of EA treatment.

The cerebellum was targeted due to its important role in the response and function of pain processing. The primary characteristic purpose of the cerebellum involves its role in motor control functions, but it has also been identified for its non-motor functioning, observed in cognitive control and learning processing [17,25,26]. The cerebellum lobules V, VI and VII were emphasized in the present study because of their functional activity. Lobule V is well-known for motor function and perception including vision, audition, general sensation and movement. Lobule VI mainly performs cogitation, and is involved in attentional system memory and reasoning. Lobule VII is associated with emotion, and communicates with the amygdala, which is critical component of analytical responses in pain and fear, and pain memory processing [27]. These three areas interactively work in pain-related circumstances, whereby the lobules V and VI principally participate in the initial pain stimulation, while lobule VII reacts to future threats and pain experiences [28]. The previous investigations mention the connection among these three lobules in pain-associated brain areas such as the thalamus, hypothalamus, amygdala, and periaqueductal gray [27]. To study inflammatory pain in the animal model research, CFA injection is one of the most commonly used methods which presents stable results in first three days, persisting for up to two weeks after the injection [6]. The murine model of CFA-inducing inflammatory pain is measured by using von Frey and Hargreaves’ tests [2]. The previous results displayed significant decreases of both withdrawal thresholds and withdrawal latency on the day after injection when contrasted with a control group [29]. This supports our similar results in terms of behavioral examinations. Furthermore, our immunoblotting and immunofluorescence results revealed the significant elevation of protein density in the cerebellum lobules V, VI and VII in the CFA group when compared to the control group, which firmly provides novel information on inflammatory pain in the cerebellum.

One of the most substantial nociceptors is TRPV1, which plays an essential role in pain disorders [30]. Capsaicin is the most TRPV1-recognized stimulus, which can activate TRPV1 receptors to initiate the calcium influx [31,32]. The receptor can be also triggered by a temperature of ~43 °C, an acidity of pH ~6, voltage, and selected ion permeability and inflammatory mediators [30,33]. These activators cause the depolarization of the membrane and further cascade signaling through the intercellular kinase pathway, protein kinase C, protein kinase A-dependence and MAPK pathways [34,35]. The distribution of TRPV1 was determined both inside and outside the nervous system, implying its importance for other impairments alongside pain sensation [36]. The spinal cord, where signals propel to the brain before returning a response and the sensory cortex, which specifically responds to pain sensations, were specially examined to illustrate the pain pathways through use of immunoblotting and RT-PCR Assay analysis [37,38,39]. Later, Jennifer C. Roberts et al. determined the comparison of TRPV1 distribution in the brain, spinal cord and DRG between the TRPV1 wild-type and knockout mice. They found a significantly high density in TRPV1 wild-type mice in most of these areas, including the dorsal horn of the spinal cord [40]. Moreover, in the cerebellum, it was claimed that TRPV1 is the most abundant among the cerebral cortex, hippocampus, hypothalamus, and cerebellum [39]. The evidence contributes to the results in the current research which demonstrated TRPV1 staining in both immunoblotting and immunofluorescence outcomes. Apart from TRPV1, other nociceptors such as ASIC3, Nav1.7, Nav1.8, RAGE, and related molecules were also investigated to corroborate their mechanisms within the cerebellum. The acid sensing ion channel subtype 3 (ASIC3) is mainly activated by acid [41,42]. Nav1.7 and Nav1.8 are voltage-gated sodium channels commonly implicated in pain states [43,44]. The receptor for advanced glycation end products (RAGE) was recently showed in its interactions with the S100 family, especially S100B, which is the best characterized calcium-binding protein of its family [45,46]. These four nociceptive receptors play at least one of the important roles in the pain and inflammation process. The associated functions and mechanisms of pain through these pain receptors were also discovered alongside the MAPK and protein kinase pathways [47,48,49]. 

TRPV1 is approved in its involvement with diverse pain mechanisms in different locations that govern pain management, as was demonstrated in novel articles that also included the treatment of EA [50,51,52]. EA is acupuncture with modern manipulation using electric stimulation at an identifiable amplitude and frequency with a constant pulse [24]. The effectiveness of EA for frequent types of pain and its related conditions via the TRPV1 nociceptor has been studied at peripheral, spinal and supraspinal levels [14]. See Yoon Seo et al. mentioned that EA side-effects in clinical studies are significantly negligible and can be avoided simply by being cautious and hygienic in treatment practice [23]. Focusing on inflammatory pain, the ST36 acupoint is commonly selected for pain therapy due to the TCM theory, which diagnoses hyperalgesia as qi stagnation and blood stasis [53]. Luis Ulloa et al. declared that EA treatment at the ST36 acupoint can inhibit inflammatory activation and nociceptive sensation by increasing mechanical stimulation through sciatic nerve involvement and the adjoining connective tissue. EA at the ST36 acupoint activates increases in the influx of potassium channels, and decelerates transient calcium channels through Adenosine A1 receptors [24,54]. Additionally, other ligands such as opioids, glutamate, γ-aminobutyric acid (GABA), 5-hydroxytryptamine (5-HT), interleukin-1β and 6 (IL1β, IL6) substantiated an association in inflammatory pain attenuation after receiving EA treatment [14].

Acupuncture therapy at the ST36 acupoint has the potential to provide a viable opportunity for illness alleviation due to its broad response through the manipulation of TRPV1 [55]. Acupuncture therapy offers numerous advantages over these traditional analgesics as it lacks many of the negative side effects associated with these conventional pain medicines, such as addiction and gastrointestinal damage [56]. In addition, EA provides a more modern and less cumbersome version of acupuncture therapy. Although the efficacy of acupuncture therapy has been attested in a number of studies, research investigating and elucidating the specific mechanism of TRPV1 in the cerebellum for the reduction of inflammatory pain remains lacking. Inflammatory pain correlates with multi-diverse notions that may partially, directly or indirectly link to the cerebellar activity. Hence, our research has the potential to verify the scientific validity acupuncture and EA methods by focusing on the TRPV1, and related pathways, in the cerebellum in order to provide a viable alternative to conventional analgesic usage in medicine. The paw withdrawal threshold and latency display significant increases in the CFA + EA and CFA + KO groups when compared to the CFA and CFA + SHAM groups, stabilizing to the baseline of the control group. Meanwhile, the data from the thermal Hargreaves’ test of CFA + KO reveal rapid recovery in contrast to other inflammatory-pain-inducing subjects due to the fact that the TRPV1 receptor is especially sensitive to the heat. Similarly, Ruth Ruscheweyh et al. declared that the cerebellum function is more prominently involved in thermal hyperalgesia than mechanical hyperalgesia [25]. The analogous tendencies of nociceptors and related protein levels were displayed in the cerebellum lobules V and VII via immunoblotting and immunofluorescence evidence. However, the results of the cerebellum total lobule VI using immunoblotting methods show non-significance, differing from the significant differences of protein density in the cerebellum lobules VIa using immunofluorescence methods, since immunoblotting techniques cannot reliably separate the two portions of lobule VI, but immunofluorescence can visibly do so. These outcomes suggest the influence of a larger visible dimension of lobule VIb compared to lobule VIa in the immunoblotting technique. In summary, the information suggests that inflammatory pain can be improved by TRPV1 functional reduction which attenuates the pain mechanisms in the cerebellum lobules V, VIa and VII. For future study, mixed data of the cerebellum lobule VI of this study could be examined with other biomolecular techniques. Accordingly, NFkB signaling is a complex pathway and correlates with various molecules. This study has verified only some of its associated connections, such as PI3K/Akt, ERK and PKC pathways. Future studies can be conducted to further assess related mechanisms of the NFkB pathway. This has been included in the limitations of this study.

Finally, we conclude that CFA-inducing inflammatory pain impacts on the protein modifications of TRPV1, ASIC3, Nav1.7, Nav1.8, RAGE, S100B, pPI3K, pmTOR, pAkt, pERK, pPKAIIα, pPKCε, pCREB and pNFkB in the cerebellum. Additionally, both 2 Hz EA at the ST36 acupoint and TRPV1 gene deletion can regulate the inflammatory pain via the nociceptors and downstream molecules in the cerebellum lobules V, VIa and VII (Figure 7).

## 4. Materials and Methods

### 4.1. Experimental Animals

Female C57/BL6 mice were already purchased from BioLasco Taiwan Ltd. (Yilan, Taiwan). The animals were housed in a Plexiglas cage with free access to standard mice chow and water ad libitum. They were placed in a temperature-controlled room (25 ± 2 °C) under a 12:12 h light-dark cycle (from 8.00 a.m. to 8.00 p.m.) with relative humidity of 60 ± 5%. The experiment started when the subjects were 8 to 12 weeks old and weighed 18 to 22 g. The experiment was first approved by the Institute of Animal Care and USE Committee of China Medical University (Permit no. 2016-061, 16/08/2019), Taiwan, following the Guide for the use of Laboratory Animals (National Academy Press). The subjects were randomly divided into 5 groups: control group, CFA pain inducing group (CFA), EA treatment group (CFA + EA), sham EA treatment group (CFA + SHAM), and TRPV1 gene deletion group (KO + CFA), with 8 mice in each group. On Day 0, all subjects were observed for the baseline of experiment before inducing inflammatory pain. The subjects were put into fixation machines under anesthesia with 1% isoflurane; the subjects in the CFA, CFA + EA, CFA + SHAM, and KO + CFA groups were injected with 20 μL CFA (Sigma, Saint Louis, MO, USA) intraplantar in the right hind paw using a 25-gauge needle, while the subjects in the control group were injected with 20 μL of normal saline. They were allowed to recover in their home cages for 24 h before continuing with the experiment. On Day 1 to Day 3, behavioral tests were conducted daily, while the EA treatment was performed on Day 2 and Day 3. The procedures were performed during the 9.00–11.00 a.m. period. Then, animals were returned to home cages after the process. None of the subjects displayed any side effects from either the CFA injection or EA treatment, and none of the animals were removed during the experimental period. The protocol was designed to sacrifice the subjects on Day 3 of experiment after EA treatment and behavior examinations had been completed. Efforts were made to minimize the number of animals used and their suffering.

### 4.2. Behavioral Examination

The subjects were put in the chamber in the behavioral room with temperature at 25 °C and allowed a 30 min habituation period before initiation of the examinations. The examinations of subjects were only conducted when they were calm without sleeping or grooming action. The electric von Frey filament instrument (North Coast Medical, Gilroy, CA, USA) was applied by increasing the pressure from underneath the right hind paw. The brisk withdrawal or paw flinching from von Frey filament test was recorded and determined as the mechanical sensitivity. The thermal sensitivity result of the Hargreaves’ test using IITC analgesiometer (IITC Life Sciences, SERIES8, Model 390 G, Woodland Hills, CA, USA) were considered based on withdrawal latency time with a 20 s cut-off time. The mechanical and thermal behavioral responsiveness were repeated three times in each subject and mean values were calculated. To avoid the conditioning to stimulation, 10 min period were interpolated in two tests.

### 4.3. EA Treatment

The subjects in CFA + EA and CFA + SHAM groups received EA treatment or placebo treatment at the bilateral sides of ST36 on Day 2 and Day 3 post-injection. The bilateral sides of ST36 were selected, then the 0.2 × 13-inch needles (0.5 in, 32 G; Yu Kuang Chem. Ind. Corp., Tainan Taiwan) were punctured at the 3 mm depth of acupoints. For subjects in CFA+EA group, the needles were attached with the electric stimulator at the amplitude of 1 mA, a frequency of 2 Hz with a constantly pulse width of 150 µs for 20 min. The intensity has to strong enough to provoke slight twitches of the limbs.

### 4.4. Immunoblotting

Immunoblotting is an analytical technique that uses specific antibody binding to a unique epitope of the interested protein. In this study, subjects were euthanized with an overdose of 5% isoflurane by inhalation before decapitation and the cerebellum lobe VI and VII were immediately dissected out. The brain samples were frozen in ice before being stored at −80 °C. Total proteins were prepared by abraded and lysed in solution of 50 mM Tris-HCl pH 7.4, 250 mM NaCl, 1% NP-40, 5 mM EDTA, 50 mM NaF, 1 mM Na3VO4, 0.02% NaN3 and 1× protease inhibitor cocktail (AMRESCO) before being centrifuged and being added with a bromophenol blue dye. Protein from each sample was loaded on 8% and 12% SDS-Tris glycine gel electrophoresis, followed by transfer onto the PVDF membrane. The membrane was blocked with 5% nonfat milk in TBS-T buffer (10 mM Tris pH7.5, 100 mM NaCl, 0.1% Tween 20) for 1 h at room temperature; afterward, it was incubated with antibodies (1:1000, Alomone, Jerusalem, Israel): anti-tubulin, anti-TRPV1, anti-Nav1.7, anti-Nav1.8, anti-ASIC3, anti-RAGE, anti-pmTOR, anti-pPI3K, anti-pAkt, anti-pERK, anti-pS100B, anti- pNFκB, anti-pCREB, anti- pPKCε or anti-pPKAIIα in TBST with 1% bovine serum albumin. Peroxidase-conjugated anti-mouse, anti-rabbit or anti-goat antibody (1:5000) was used as a secondary antibody. The protein bands on membranes were visualized by an enhanced chemiluminescencent substrate kit (PIERCE, Rockford, IL, USA) with LAS-3000 Fujifilm (Fuji Photo Film Co. Ltd., Tokyo, Japan). The image densities of the specific bands were quantified using NIH ImageJ software (Bethesda, MD, USA).

### 4.5. Immunofluorescence

Animals were euthanized with a 5% isoflurane via inhalation and intracardially perfused with normal saline followed by 4% paraformaldehyde. The brain was immediately dissected and post fixed with 4% paraformaldehyde at 4 °C for 3 days. The tissues were placed in 30% sucrose for cryoprotection overnight at 4 °C. The brain was embedded in Optimal cutting temperature (OCT) compound and rapidly frozen using liquid nitrogen before storing the tissues at −80 °C. Frozen segments were cut at 20-μm width on a cryostat then instantaneously placed on glass slides. The samples were fixed with 4% paraformaldehyde, then incubated with blocking solution, consisting of 3% BSA, 0.1% Triton X-100, and 0.02% sodium azide, for 1 h at room temperature. After blocking, the samples were incubated with the primary antibody (1:200, Alomone, Jerusalem, Israel), TRPV1, pPKAIIα, pPI3K, and pCREB, prepared in 1% bovine serum albumin solution at 4 °C overnight. Afterward, the samples were incubated with the secondary antibody (1:500), 488-conjugated AffiniPure donkey anti-rabbit IgG (H + L), 594-conjugated AffiniPure donkey anti-goat IgG (H + L) and Peroxidase-conjugated AffiniPure donkey anti-mouse IgG (H + L) for 2 h at room temperature before being fixed with cover slips for immunofluorescence visualization. The samples were observed by an epi-fluorescent microscope (Olympus, BX-51, Tokyo, Japan) with 20× numerical aperture (NA = 0.4) objective. The images were analyzed by NIH ImageJ software (Bethesda, MD, USA).

### 4.6. Data Analysis

Statistical analysis was performed using the SPSS statistic program and figures were drawn using the Lab Origins program. All data were presented with the mean ± standard errors. Statistical comparisons were evaluated using the ANOVA test following by Tukey’s post hoc test to determine the significance of differences. Values of *p* < 0.05 were considered as significant.

## Figures and Tables

**Figure 1 ijms-21-03312-f001:**
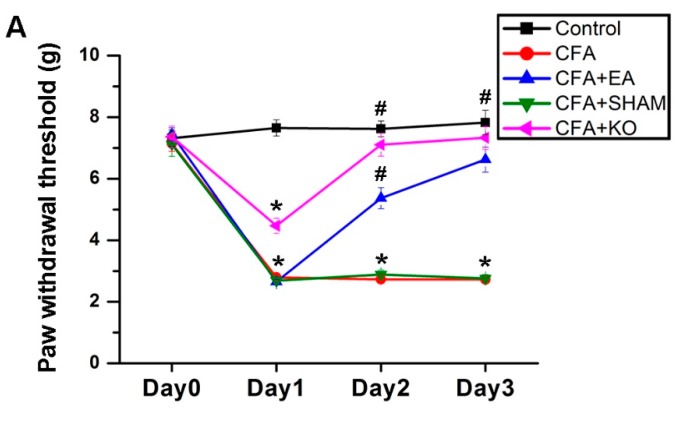
Comparative graph of paw withdrawal threshold and latency of the Complete Freund’s Adjuvant (CFA)-induced inflammatory pain subjects after Electroacupuncture (EA) treatment and transient receptor vanilloid member 1 (TRPV1) gene deletion. *n* = 8 for five groups: control, CFA, CFA + EA, CFA + SHAM, and CFA + KO underwent (**A**) mechanical von Frey and (**B**) thermal Hargreaves’ tests. The results demonstrated parallel tendencies between the two tests. The CFA injection significantly increases hyperalgesia in mechanical and thermal sensitivity. The significant reducing threshold and latency were increased using 2 Hz EA at ST36 and TRPV1 gene deletion. * *p* < 0.05 means when compared with the baseline of control group. # *p* < 0.05 means when compared with the CFA and CFA + SHAM groups.

**Figure 2 ijms-21-03312-f002:**
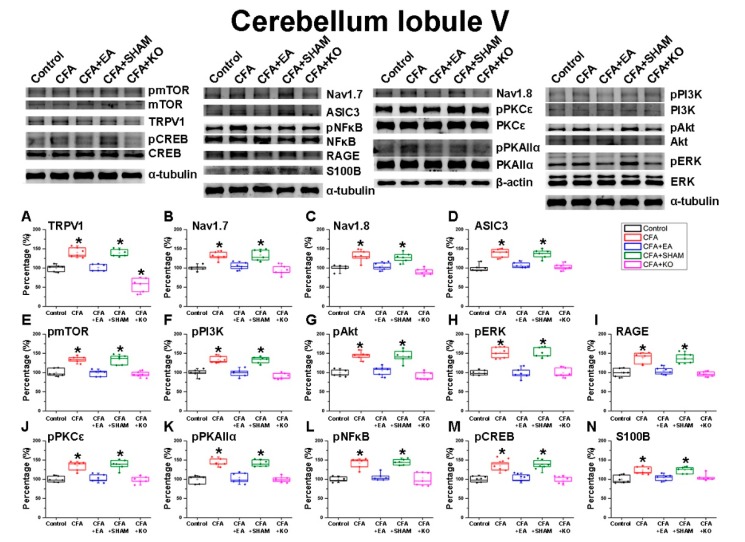
The expression levels of pain receptors and involved molecules in the cerebellum lobule V. The immunoblotting images contain five lanes of protein in the following order: control, CFA, CFA + EA, CFA + SHAM, and CFA + KO groups. There are significant decreases of (**A**) TRPV1, (**B**) Voltaged-gated Sodium (Nav)1.7, (**C**) Nav1.8, (**D**) acid sensing ion channel subtype 3 (ASIC3), (**E**) pmTOR, (**F**) pPI3K, (**G**) pAkt, (**H**) pERK, (**I**) receptor for advanced glycation end products (RAGE), (**J**) pPKCε, (**K**) pPKAIIα, (**L**) pNFκB, (**M**) pCREB, and (**N**) S100B protein level in CFA + EA and CFA+KO groups compared to CFA and CFA + SHAM groups (* *p* < 0.05).

**Figure 3 ijms-21-03312-f003:**
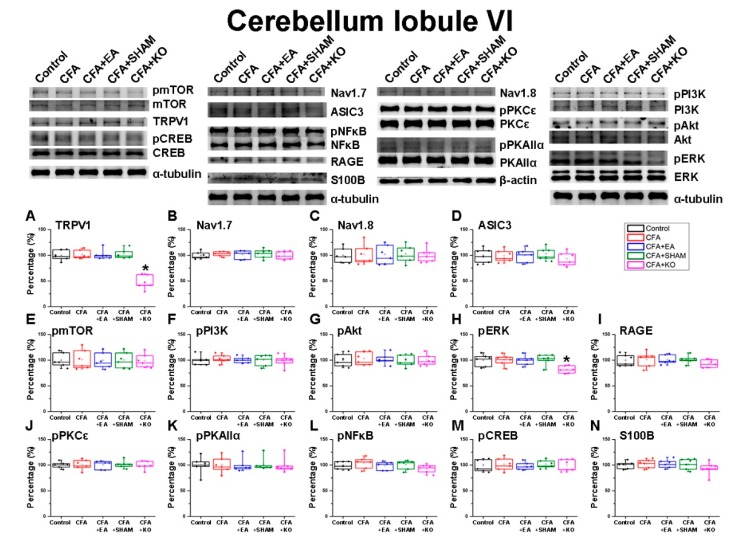
The expression levels of pain receptors and involved molecules in the cerebellum lobule VI. The immunoblotting images contain five lanes of protein in the following order: control, CFA, CFA + EA, CFA + SHAM, and CFA + KO groups. There are no significant difference for (**B**) Nav1.7, (**C**) Nav1.8, (**D**) ASIC3, (**E**) pmTOR, (**F**) pPI3K, (**G**) pAkt, (**I**) RAGE, (**J**) pPKCε, (**K**) pPKAIIα, (**L**) pNFκB, (**M**) pCREB, and (**N**) S100B protein level among the five groups of subjects (*p* > 0.05). However, the protein density of (**A**) TRPV1 and (**H**) pERK reveal significant decreases in the CFA + KO group compared to the other four wild-type groups (* *p* < 0.05).

**Figure 4 ijms-21-03312-f004:**
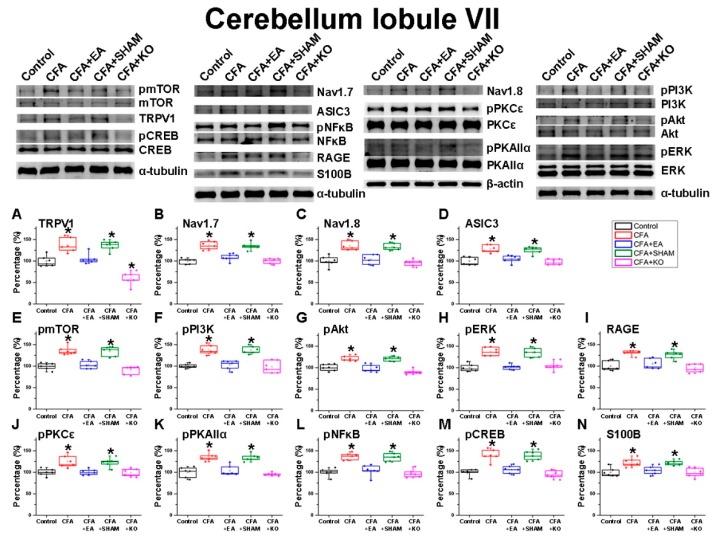
The expression levels of pain receptors and involved molecules in the cerebellum lobule VII. The immunoblotting images contain five lanes of protein in following order: control, CFA, CFA + EA, CFA + SHAM, and CFA + KO groups. There are significant decreases of (**A**) TRPV1, (**B**) Nav1.7, (**C**) Nav1.8, (**D**) ASIC3, (**E**) pmTOR, (**F**) pPI3K, (**G**) pAkt, (**H**) pERK, (**I**) RAGE, (**J**) pPKCε, (**K**) pPKAIIα, (**L**) pNFκB, (**M**) pCREB and, (**N**) S100B protein level in CFA + EA and CFA + KO groups compared to CFA and CFA + SHAM groups (* *p* < 0.05).

**Figure 5 ijms-21-03312-f005:**
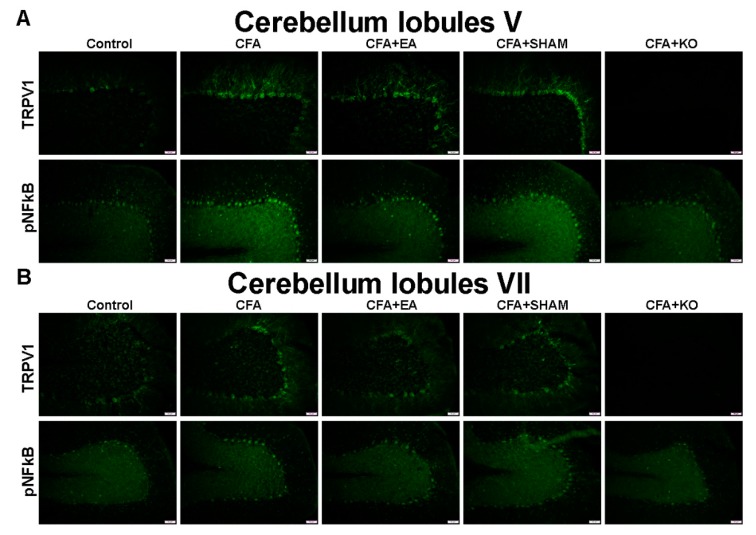
Immunofluorescence staining of TRPV1 and pNFkB protein expression in the cerebellum lobules V and VII. There are 5 subject groups: control, CFA, CFA + EA, CFA + SHAM, and CFA + KO. The effectiveness of EA treatment involves significant reduction of TRPV1 and pNFkB densities in the cerebellum lobules (**A**) V and (**B**) VII. Scale bar is 50 μm.

**Figure 6 ijms-21-03312-f006:**
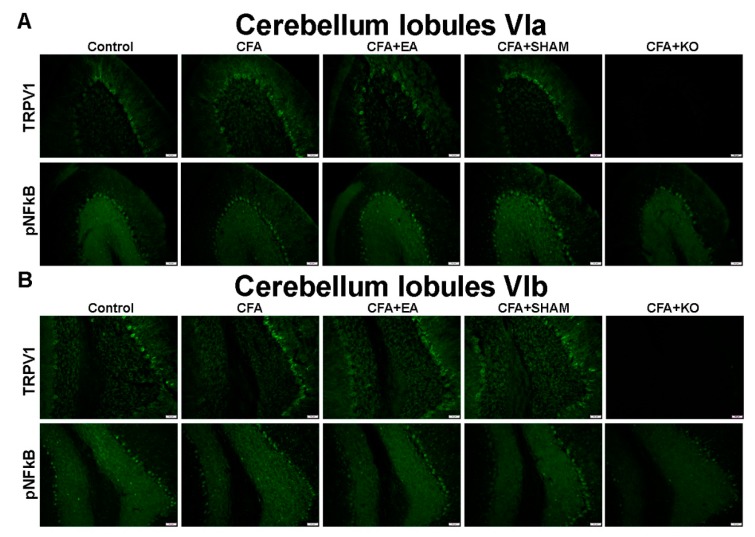
Immunofluorescence staining of TRPV1 and pNFkB protein expression in the cerebellum lobules VIa and VIb. There are 5 subject groups: control, CFA, CFA + EA, CFA + SHAM, and CFA + KO. (**A**) The effectiveness of EA treatment involves significant reduction of TRPV1 and pNFkB densities in the cerebellum lobule VIa. (**B**) In contrast, there was no nsignificant variance of TRPV1 and pNFkB density in the cerebellum lobule VIb the among five groups of subjects. Scale bar is 50 μm.

**Figure 7 ijms-21-03312-f007:**
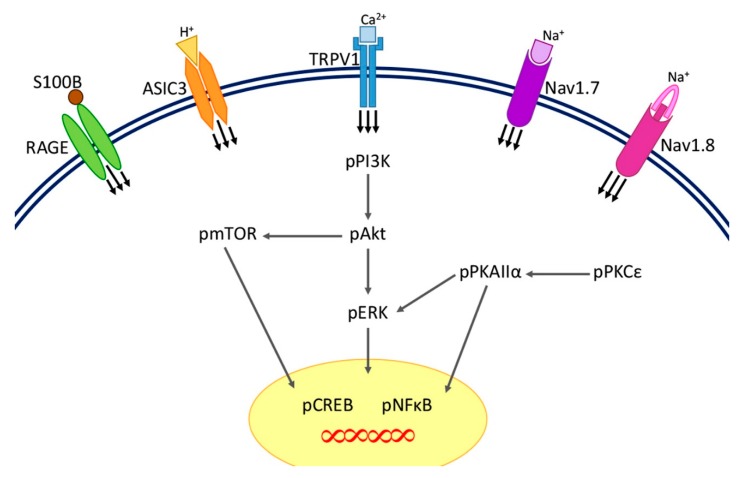
Illustration of inflammatory pain pathways attenuated by EA in the cerebellum. We found that EA at 1 mA, 2 Hz for 20 min and TRPV1 gene deletion can diminish the expression of TRPV1, Nav1.7, Nav1.8, ASIC3, and RAGE nociceptors in the CFA-induced inflammatory pain model in CFA + EA and CFA + KO groups. Furthermore, the reductions cause decreases in the signals of pPI3K, pmTOR, pAkt, pERK, pPKCε, pPKAIIα, pNFkB, pCREB, and S100B under conditions of inflammatory hyperalgesia.

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
