# Peer review of "TRPV1 Responses in the Cerebellum Lobules V, VIa and VII Using Electroacupuncture Treatment for Inflammatory Hyperalgesia in Murine Model"

_ijms, 2020, doi:10.3390/ijms21093312_

Round 1

Reviewer 1 Report

Ok for publication with this version

Author Response

Thank you for your tolerant rereading of the revised manuscript and clarification through your comments.

Reviewer 2 Report

Nice paper but I do worry that figures like 3 and 4 will be too small upon publication. Hopefully a file will be able to be downloaded like a power point for a high resolution figure.

Author Response

RESPONSE: Thank you for your valuable consideration. IJMS formatting requires figure insertion following the first mentioned in a manuscript, this might probably cause the issue you declared. However, for these figures, 300 dpi files are separately provided with the manuscript to the journal. We can also provide the original image to IJMS product team.

Reviewer 3 Report

Thank you for your revision. I believe with those further references make the background more informative and add to the significance of the discoveries. I suggest the authors include such references into the paper.

Regarding my comments:

2. I was hoping the authors could immunolabel some of the other TRP channels, so the readers understand the local relationship between these channels, also to examine if those channels are up or downregulated with treatments, with either PCR or immunostaining quantification.

3. I was hoping the authors could use a MAP2, or beta tubulinIII to demonstrate the relationship between the channels and neurons. Excuse for the unclearness. 

Author Response

  1. Thank you for your revision. I believe with those further references make the background more informative and add to the significance of the discoveries. I suggest the authors include such references into the paper.

RESPONSE: Thank you for the recommendation. The finding related to EA frequency has been inserted in the introduction section of the revised manuscript (page 3 of 23, line 67 – 75, highlighted in yellow).

  1. I was hoping the authors could immunolabel some of the other TRP channels, so the readers understand the local relationship between these channels, also to examine if those channels are up or downregulated with treatments, with either PCR or immunostaining quantification.

RESPONSE: Thank you for the comment. Due to these novel findings, we plan to further observe changes in other receptors that are either associated or unrelated to TRP channels and molecules in the cerebellum area in the near future.

  1. I was hoping the authors could use a MAP2, or beta tubulinIII to demonstrate the relationship between the channels and neurons. Excuse for the unclearness.

RESPONSE: Thank you for your precise explanation. The MAP2 and beta tubulin III neuronal markers can be included in future research along with your second piece of advice. Hopefully, the results will be examined in the future.

This manuscript is a resubmission of an earlier submission. The following is a list of the peer review reports and author responses from that submission.

Round 1

Reviewer 1 Report

It's a very interesting to look at the underlying mechanism of how electroacupuncture effect on inflammatory pain, in which both electroacupuncture at the ST36 acupoint and the deletion of TRPV1 gene can regulate the inflammatory pain via the nociceptors and downstream molecules inthe cerebellum lobules V, VIa and VII. In this paper, the authors had used different methods to check the gene expression. I would like to ask few questions, which would be great if it can be addressed.

It would be great if the authors could show quantification about protein expression level, which could use western-blot methods. It would be great to have the abbreviations show in the first place, such as EA (electroacupuncture) could be shown in the first page, instead of in the end of article.

Reviewer 2 Report

I would propose defining “ EA” in the title as well as in the abstract. It is a key focus of the paper.

line 61. The location of Zusanli acupoint should be defined, As far I found on the internet is –“Zusanli (Chinese: ST36) is an acupoint, a point of the skin that is stimulated, with various techniques, in the practice of acupuncture. It is located below the knee, on the tibialis anterior muscle, along the stomach meridian.” (Wikipedia- should use a primary source )

 Maybe later it is defined in paper but should be defined in Introduction as a reader will want to know,

Line 69 “They also contain voltage-gated Ca2+ channels that can be triggered by depolarization from climbing or parallel fibers in the cerebellum, and intracellular Ca2+ store release” …

Maybe just define the multiple types of the neurotransmitters which communicate with Purkinje cells as they have more than Ca2+ channels.

Line 79 define “CFA”

“Complete Freund's Adjuvant (CFA) consists of heat-killed Mycobacterium tuberculosis in non-metabolizable oils (paraffin oil and mannide monooleate)”

Should use a primary source.

https://en.wikipedia.org/wiki/Freund%27s_adjuvant

line 356:

 “then the 0.2 × 13-inch needles (0.5 in, 32 360 G; Yu Kuang Chem. Ind. Corp., Taiwan) were punctured at the 3 millimeters depth of acupoints. For subjects in CFA+EA group, the needles were attached with the electric stimulator at the amplitude of 1 mA, a frequency of 2 Hz with a constantly pulse width of 150 μs for 20 minutes. The intensity has to strong enough to provoke slight twitches of the limbs.”

Maybe explain why using needles is necessary as it has been shown that surface electrical stimulation does just as well in causing endogenous endorphin release.

This bring me to Line 43: “However, the recommendation of opioids as a first line treatment for chronic pain was changed because of the large number of reported side effects and risks [6, 7].”

Is this not the same but causing an endogenous release of endogenous endorphin release as taking opioids?  People do become addicted to endogenous endorphin release such as “Runner’s high” and even suggested to be related to alcohol addiction.

But it would be good to explain and it is important that EA does cause release of endogenous endorphin.

But also that opiate alkaloid drugs, such as morphine, is different in the effects than release of endogenous endorphin (https://www.cell.com/neuron/fulltext/S0896-6273(18)30329-5).

There will be a question from any reader. Does it matter if one stimulated say the front legs (of rodents), back muscles or even just under the skin in the belly? Any control for locations of electric stimulation?

Small notes:

Define “EA” in title or in abstract – line 59 Define “CFA” – line 13 Define “SHAM” – line 16 Over Expression is 1 word (overexpression) – line 21 Sentence “Even though” does not make logical sense. Seems like there is more to be added to the end of the sentence. Maybe consider combining the two sentences – line 41-43 Define NSAIDs and maybe list example – line 44 If you are going to use past tense here (line 77) then I think you should use past tense in abstract (line 20) ex. Reveals to revealed Possibly explain how you came to hypothesis that alterations in the cerebellum under inflammatory pain could be relieved using ST36 acupoint and not another acupoint – line 79 “Wuing” not a word. Did you mean “using the” – line 179

Reviewer 3 Report

The authors aim to draw the conclusion that TRPV1 in cerebellum specific regions respond to EA treatment and improve the mouse models of inflammatory hyperalgeasia (IH). Several questions or tests could be addressed right away, higher EA can damage or rescue even more IH? I assume different disease models require different frequency or length of time for treatment. Are there other TRP channels or other nociceptors localized in the cerebellum V regions? Could you use another marker stain as a reference for TRPV1? What about other mechanisms related to NFkb pathway? If the authors already have the data like this, they should be included in the manuscript.